# The characteristic of patulous eustachian tube patients diagnosed by the JOS diagnostic criteria

Yoshinobu Kawamura[1,2], Ryoukichi Ikeda[1,2]\*, Toshiaki Kikuchi[1], Hiromitsu Miyazaki[1,2], Tetsuaki Kawase[1], Yukio Katori[1], Toshimitsu Kobayashi[1,2]

**1** Department of Otolaryngology-Head and Neck Surgery, Tohoku University, Graduate School of Medicine, Sendai, Miyagi, Japan, **2** Sen-En Rifu Otologic Surgery Center, Rifu, Miyagi, Japan

\* ryoukich@hotmail.com

**Data Availability Statement:** All relevant data are in the paper and supporting information files.

**Funding:** The study was supported by a grant from JSPS KAKENHI Grant Number 18K09366 and

## Abstract

### Objective

The objective is to describe characteristics of patients diagnosed with patulous Eustachian tube (PET) using the Diagnostic Criteria proposed by Japan Otological Society, and to evaluate the efficiency of objective tests to determine patent Eustachian tube.

### Study design

Retrospective.

### Setting

Tertiary referral center.

### Subjects

A retrospective survey of medical records in Sen-En Rifu Hospital identified 78 ears of 56 patients with "Definite PET" diagnosed by the JOS Diagnostic Criteria between January 2017 and December 2017.

### Method

Initial diagnosis, aural symptoms (voice autophony, aural fullness and breathing autophony), tubal obstruction procedures (posture change and pharyngeal orifice obstruction) and objective findings (tympanic membrane movement, Tubo-Tympano-Aerodynamic Graphy (TTAG) and sonotubometry) were evaluated. In addition, sonotubometry with postural change (Ohta's method), sitting CT and a newly devised PHI-10 score were also examined.

### Results

Voice autophony, aural fullness, and breathing autophony were observed in 93.6%, 87.2%, 78.2%, respectively. In 91% of the ears, PET symptoms improved by postural change from sitting to the lying / forward-bending position. Synchronous movement of the TM upon

18K16872. The funders had no role in study design, data collection and analysis, decision to publish, or preparation of the manuscript.

**Competing interests:** The authors have declared that no competing interests exist.

respiration was observed in 69.1% of the ears. Positive findings of TTAG were observed in 75.6% of ears. Positive findings of sonotubometry were found in 55.1% of ears. Sonotubometry with postural change (Ohta's method), when the cut-off value of over 10dB was used, was positive in 45.2% of ears. Newly devised PHI-10 score representing severity of subjective symptoms classifying patients into no handicap, mild handicap, moderate handicap and severe handicap were observed in 12.2%, 10.8%, 18.9% and 58.1% of ears, respectively. The evaluation of the extent of patency of the ET by sitting CT indicated completely open, closed-short, and closed-long, in 68.6%, 11.4% and 21.4% of ears, respectively. Compared to the closed group, the completely open group had a significantly higher frequency of positive breathing autophony, positive sonotubometry, and positive Ohta's method.

## Conclusion

The characteristics of main symptoms and the efficiency of various tests in PET diagnosis were analyzed based on data obtained from "Definite PET" patients diagnosed by the JOS Diagnostic Criteria. The greater the availability of tests to evaluate PET, the greater the opportunities to diagnose "Definite PET". In particular, tests measuring pressure transmission between the nasopharynx and middle ear, such as TM observation and TTAG, are more sensitive than sonotubometry measuring sound transmission.

## Introduction

The Eustachian tube (ET) is normally closed but opens temporarily to fulfill a diverse range of functions such as ventilation, clearance and protection of the middle ear cavity. Patulous Eustachian tube (PET) patients suffer from symptoms such as aural fullness and autophony of voice or breathing sounds due to an abnormally open ET [1, 2]. The common cause of PET is weight loss [3, 4]. Other causes of PET include pregnancy, oral contraceptives [2], radiation therapy, sectioning of the trigeminal nerve [5], tonsillectomy, and adenoidectomy. PET patients are usually observed to have tympanic membrane (TM) movements during ipsilateral nasal breathing. To diagnose PET, several objective and subjective findings, such as medical

**Table 1. The diagnostic criteria for patulous eustachian tube proposed by Japan Otological Society (JOS).**

| The diagnostic criteria of PET by the Japan Otological Society |
| --- |
| 1. There are subjective symptoms |
| One or more of the following symptoms included: voice autophony, a sense of aural fullness, and breathing autophony |
| 2. Tubal obstruction procedures (A or B) clearly improves symptoms |
| A. Posture change to the lying / lordotic position |
| B. Pharyngeal orifice obstruction treatment (swab, gel, etc.) |
| 3. There is at least one of the following objective findings of a patent E-tube: |
| A. Respiratory fluctuation of the tympanic membrane |
| B. Variations of external auditory meatus pressure synchronized with nasopharyngeal pressure |
| C. The sonotubometry shows (1) the test tone sound pressure level less than 100 dB or (2) an open plateau pattern. |
| If all three criteria are met (1+2+3), the diagnosis is "Definite PET", whereas if only two criteria are met (1+2 or 1+3), the diagnosis is "Possible PET". |

history, physical examination and ET function tests, are combined because there is no single test available to evaluate ET function accurately [6, 7]. For this reason, each institution defined PET according to their own criteria and widely accepted diagnostic criteria for PET has not been established until recently, when the Otological Society of Japan (JOS) Diagnostic Criteria for Patulous Eustachian Tube was published (Table 1). This criteria use the terms "Definite PET" and "Possible PET" [7], where "Definite PET" is defined as cases of PET with 100% certainty and "Possible PET" is defined as cases with possibility but less certainty. Although JOS has developed the useful criteria in the diagnosis of PET, there has not been any report documenting the characteristics of patients diagnosed by the criteria.

This study was conducted to investigate the usefulness of this criteria by exploring the characteristic of patients diagnosed as "Definite PET" by JOS Diagnostic Criteria for PET.

## Materials and methods

### Clinical examination

Criteria of 1 and 2A were detected by diagnostic interview (Table 1).

Criteria of 2B was conducted when 2A was negative.

Criteria of 3A was detected using otomicroscopy and endoscopy in sitting position. The TM movements during ipsilateral nasal breathing was defined as positive.

Criteria of 3B was detected by Tubo-Tympano-Aerodynamic Graphy (TTAG) [8]. The TTAG and sonotubometry were performed using a commercially available machine (JK05A; Rion, Tokyo, Japan). Pressure changes in the external auditory canal (EAC) and the nasopharynx were simultaneously recorded using the manometry mode of the TTAG. Positive findings of TTAG were defined as an EAC pressure change synchronous with that in the ipsilateral nasopharynx [9], and these findings reflect the movement of the TM upon respiration or sniffing (Fig 1).

Criteria 3C used sonotubometry [10]. Sonotubometry automatically creates the input sound pressure level (SPL) whereby the acoustic signal comprises a 7 kHz octave band noise at the nostril, which enables pre-set level 50 dB SPL output in the EAC [11]. Positive findings of PET were defined as a lowering of probe tone SPL to below 100 dB (Fig 2 left) or a so-called "open plateau pattern" obtained when the ET opens upon swallowing and remains open thereafter (Fig 2 right).

### Subjective PET symptoms evaluation scales

The patulous Eustachian tube handicap inventory-10 (PHI-10) scale was devised to evaluate the severity of subjective PET symptoms [12] (Table 2).

### Sonotubometry with postural change (Ohta's method)

Sonotubometry with postural change (Ohta's method) was previously reported as follows [11, 13]. Sonotubometry with the postural change from the forward-bending to the sitting positions evaluate the change in sound pressure transmitted from nasopharynx in monitored level at the EAC during the postural change. The acoustic transfer function via the ET was compared in the sitting and forward-bending positions (Fig 2B). Changes in probe tone SPL exceeding 10dB was regarded as a positive finding of PET.

### Morphologic evaluation by sitting 3-D CT

The 3-D cone-beam computed tomography (CBCT) (Accuitomo; Morita, Kyoto, Japan) in the sitting position was used as previously reported [14–16]. The multiplanar reconstruction

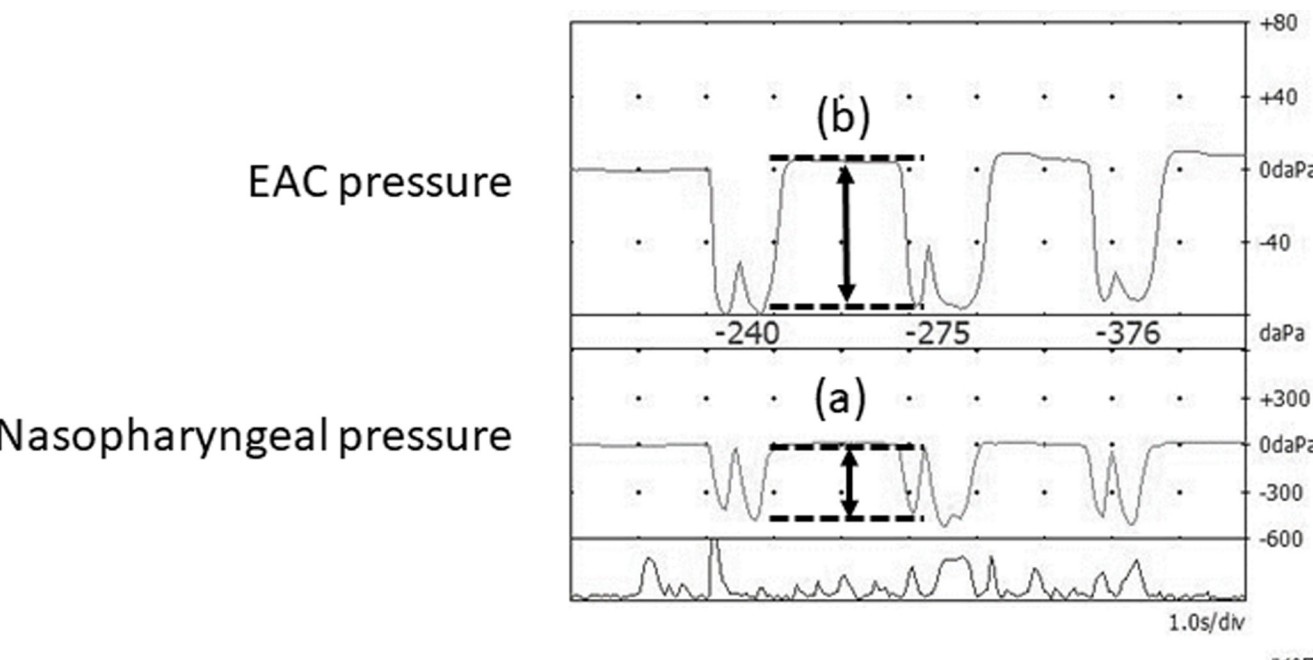

**Fig 1. A typical example of TTAG measurement in a case of PET.** The TTAG can objectively record synchronous changes in the pressure induced by movement of the tympanic membrane upon respiration or sniffing. Pressure changes were evaluated as pressure transmission ratio: (pressure b: EAC pressure) / (pressure a: pharyngeal pressure). EAC indicates external auditory canal; TTAG, tubotympanoaerodynamography.

(MPR) technique was used to reconstruct 1-mm-thick gapless images, parallel and perpendicular to the ET long axis. The opened section of the ET lumen was revealed as a hyperlucent area. The closed section of ET lumen was measured and ears were assigned according to their open length of the ET, to one of three groups as follows: completely open group (Fig 3. left), closed-short (3mm or less) group (Fig 3. middle), and closed-long (longer than 3mm) group (Fig 3. right).

## Statistical analysis

Summary statistics were performed for patient demographics, symptoms, methods of diagnosis for PET, and clinical examination findings.

Mann-Whitney's U test was performed using SPSS version 20 (IBM, Chicago, IL, USA). Differences with a corrected p-value of less than 0.05 were considered significant. Data are presented as mean ± standard deviation.

All procedures of the present study were approved by the ethical committee of Sen-En Hospital Institutional Review Board (IRB). All parts of the present study were performed in accordance with the guidelines of the Declaration of Helsinki (1991).

## Results

### Patients

A prospective survey of medical records in Sen-En Rifu Hospital identified 56 patients, (21 male and 35 female subjects aged 12 to 88 years, average 49.3±19.0 years), 78 ears (bilateral ear: 22 cases, right ear: 14 cases and left ear: 20 cases) with definitive PET between January 2017 and December 2017.

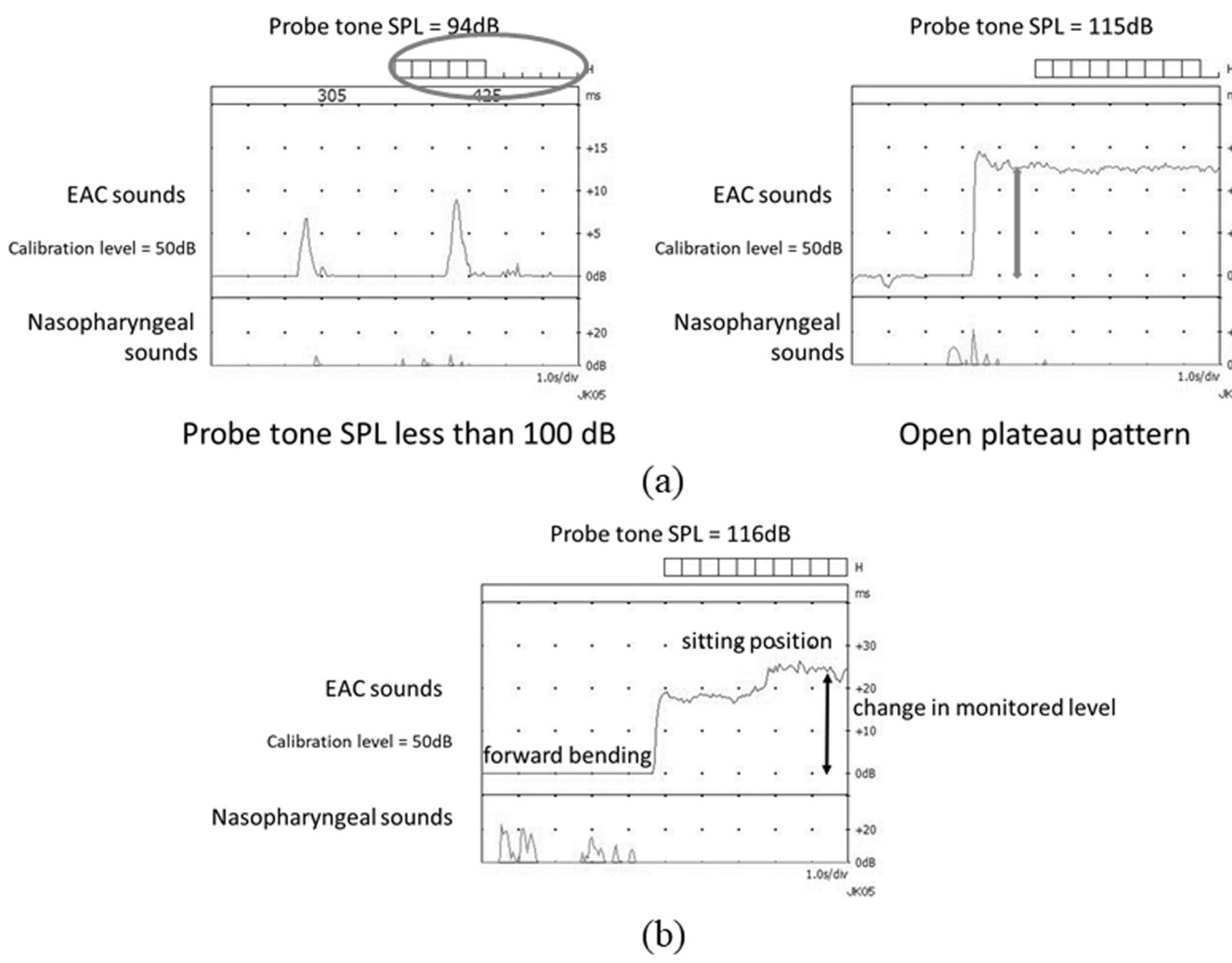

**Fig 2. (a) Typical examples of sonotubometric measurements in cases of PET.** Lowering of the probe tone SPL to less than 100dB (Left: circle). The ET opens when swallowing and remains continuously open thereafter (Right: gray arrows). ET indicates Eustachian tube; SPL, sound pressure level. **(b) A typical example of sonotubometric measurements with postural change in a case of PET.** Sound attenuations from the speaker to the microphone in the sitting and forward-bending positions. The level difference was observed as a dynamic change of probe tone SPL in response to the postural change from the forward-bending to sitting positions. EAC indicates external auditory canal.

Table 2. Patulous Eustachian tube handicap inventory-10 (PHI-10).

| No | Question | yes: 4 | sometimes: 2 | no: 0 |
|----|----------|--------|--------------|-------|
| 1 | Because of your symptom is it difficult for you to concentrate? | | | |
| 2 | Does the loudness of your symptom make it difficult for you to hear people? | | | |
| 3 | Does your symptom make you angry? | | | |
| 4 | Do you feel as though you cannot escape your symptom? | | | |
| 5 | Does your symptom interfere with your ability to enjoy social activities? | | | |
| 6 | Because of your symptom do you feel frustrated? | | | |
| 7 | Does your symptom interfere with your job or household responsibilities? | | | |
| 8 | Do you feel that your symptom has placed stress on your relationships with members of your family and friends? | | | |
| 9 | Do you find it difficult to focus your attention away from your symptom and on to other things? | | | |
| 10 | Does your symptom make you feel anxious? | | | |

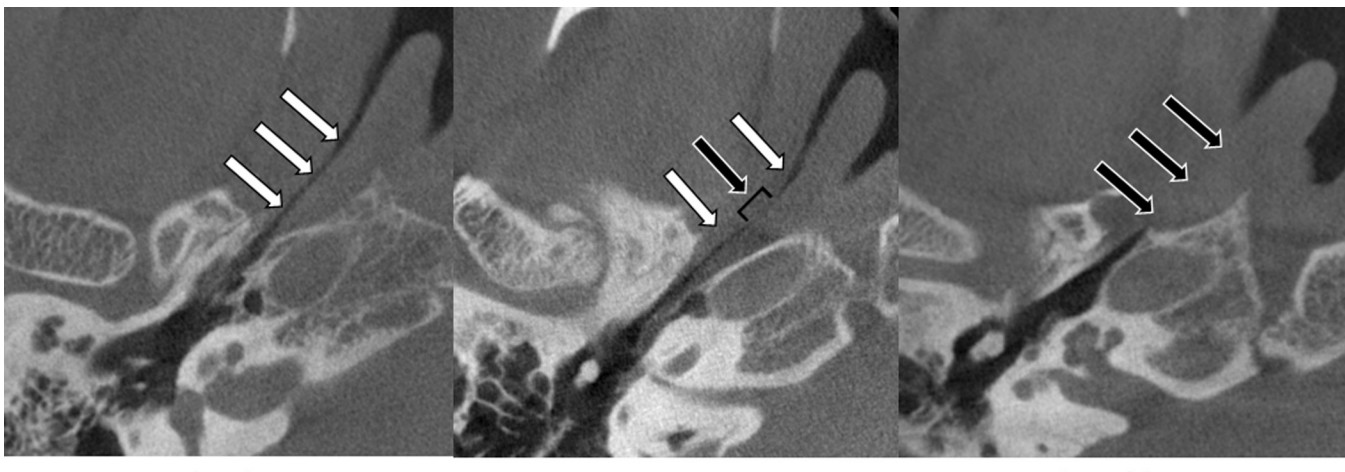

**Fig 3. Representative CT images (axial view) in patients with PET.** Left: completely open. Middle: closed-short (3mm or less). Right: closed-long (longer than 3mm). White arrows indicate ET open. Black arrows indicate ET closed. CT indicates computed tomography; ET, Eustachian tube.

## Timing of diagnosis as "Definite PET"

Seventy-one of 78 ears (91.0%) were diagnosed as "Definite PET" at the first consultation in our department (Table 2). The remaining ears were initially diagnosed as "Possible PET" and diagnosed as "Definite PET" at following visits.

**Table 3. Summary of the clinical features of patients.**

| | | Total | | |
|---|---|---|---|---|
| | | Total | Positive | % |
| Diagnosis of PET at first consultation | | 78 | 71 | 91.0% |
| Aural symptoms | | | | |
| | Voice autophony | 78 | 73 | 93.6% |
| | Aural fullness | 78 | 68 | 87.2% |
| | Breathing autophony | 78 | 61 | 78.2% |
| Tubal obstruction procedures | | | | |
| | Posture change | 78 | 71 | 91.0% |
| | Pharyngeal orifice obstruction | 7 | 7 | 100.0% |
| Objective findings of patent E-tube | | | | |
| | TM movement | 68 | 47 | 69.1% |
| | TTAG | 78 | 60 | 76.9% |
| | Sonotubometry | 78 | 43 | 55.1% |
| | below 100dB | 78 | 35 | 44.9% |
| | plateau type | 78 | 11 | 14.1% |
| Other objective findings of patent E-tube | | | | |
| | Ohta method | | | |
| | upper 10dB | 62 | 28 | 45.2% |
| | Sitting 3-D CT | | | |
| | completely open | 70 | 48 | 68.6% |

## Subjective findings

Voice autophony, a sense of aural fullness, and breathing autophony were observed in 73 (93.6%), 68 (87.2%), 61 (78.2%) ears, respectively. Fifty-three (67.9%) ears had all the three symptoms (Table 3).

## Tubal obstruction procedures

Seventy-one (91.0%) ears reported improvement of PET symptoms by postural change from sitting or upright to lying or forward-bending position (Table 3). In the remaining 7 ears, the PET symptoms were remarkably alleviated by pharyngeal tubal orifice obstruction treatment.

## Objective findings

Respiratory fluctuation of the TM was observed in 47 of 68 ears (69.1%). Positive findings of TTAG were observed in 60 of 78 (76.9%) (Fig 4A). Positive findings of sonotubometry, a probe tone SPL less than 100 dB, was found in 35 of 78 ears (44.9%) (Fig 4B) and an open plateau pattern in 11 of 78 ears (14.1%), respectively (Table 3). As three ears were positive for both probe tone testing and an open plateau pattern, 43 of 78 ears (55.1%) were judged positive in sonotubometry. In sixty-eight ears in which all the three tests (TM movement, TTAG, sonotubometry) were conducted, all three were positive in 20 of 68 ears (29.4%) (Fig 5). In the 68 ears, TM movement, TTAG and sonotubometry were observed as single positive objective findings in 13.2%, 16.2% and 1.5%, respectively (Fig 5).

## Sonotubometry with postual change (Ohta's method)

Sixty-two ears were assessed by Ohta's method (Fig 4C). When the cut-off value of this method was defined as a probe tone SPL exceeding 10 dB, 28 (45.2%) were positive (Table 3).

## Subjective PET symptoms evaluation scales

Seventy-four ears were evaluated by PHI-10. No handicap (0–8), mild handicap (10–16), moderate handicap (18–24) and severe handicap (26–40) were observed in 9 (12.2%), 8 (10.8%), 14 (18.9%) and 43 (58.1%), respectively (Fig 4D).

## Sitting 3-D CT

Findings of the sitting 3-D CT in 70 ears were evaluated and classified into completely open group, closed-short (3mm or less) group, and closed-long (longer than 3mm) group, and each group consisted of 48 (68.6%) (Table 2), 8 (11.4%) and 14 (20.0%) ears, respectively. The incidence of breathing autophony, positive findings of sonotubometry and Ohta's method was significantly higher in the completely open group than closed group (Table 4).

## Discussion

In this study, we analyzed characteristics of patients diagnosed as "Definite PET" in JOS Diagnostic Criteria for PET. A patient with PET is best diagnosed through a well-structured examination including patient history, physical examination with thorough observation of movements of the TM and objective findings using several testing equipment [1, 17].

## Timing of diagnosis as "Definite PET"

The JOS Diagnostic Criteria for PET was defined to avoid any contamination of "Definite PET" with uncertain cases, so that "Definite PET" accurately reflects PET [7]. Possible PET

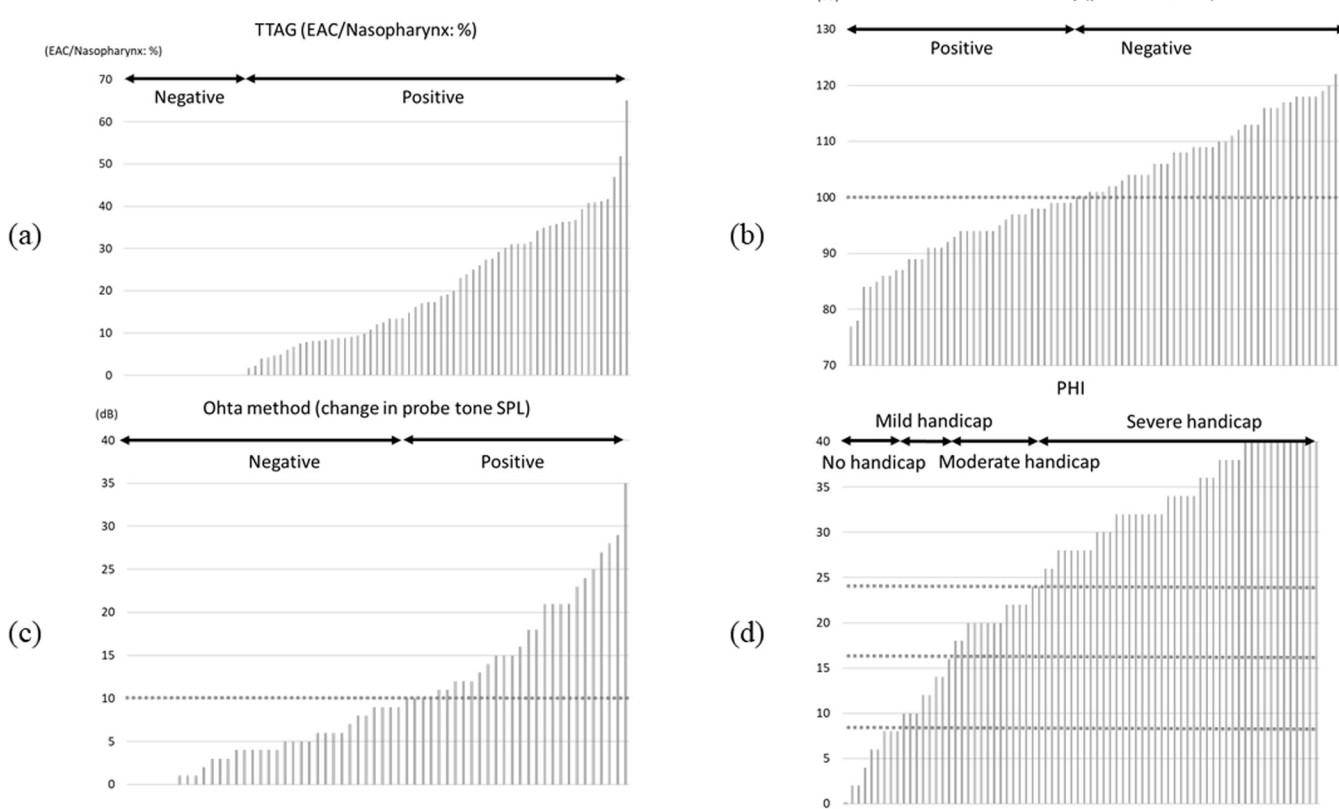

**Fig 4. (a) Summary of results from TTAG.** The vertical axis indicates the ratio of EAC / Nasopharynx: %. EAC indicates external auditory canal. **(b) Summary of results from sonotubometry.** The vertical axis indicates the probe tone SPL (dB). Positive results were found in 55.1%. **(c) Summary of findings from Ohta's method.** The vertical axis indicates the change of probe tone SPL in response to the postural change from the forward-bending to sitting positions (dB). Positive results were found in 45.2%. **(d) Summary of results from PHI-10.** The vertical axis indicates the score of PHI-10.

was intended to minimize the number of cases that could be accidentally excluded even in the presence of some suspected findings because most patients report that their PET symptoms are intermittent, even in severe cases [6].

In this study, 71 out of 78 (91%) ears were diagnosed as "Definite PET" at the first consultation and only 7 ears (9%) required more than one visit before reaching the diagnosis of "Definite PET". Such high incidence of initial accurate diagnosis rate may be due to the fact that our institute received many referrals of intractable PET from other clinics from all over Japan.

## Posture change

Bothersome PET symptoms are usually relieved by posture change to the lying or forward-bending position. Ward et al. reported that 65.3% of patients experienced relief with the head in a dependent position [18]. However, in some cases, PET symptoms did not improve by these posture changes and the TM movements were confirmed even in the recumbent position. The JOS Diagnostic Criteria recommend pharyngeal orifice obstruction treatment using swab, gel, etc. in order not to miss these cases. In this study, seven cases (9.0%) needed pharyngeal orifice obstruction treatment to confirm the diagnosis of "Definite PET", because they did not report improvement in symptoms by postural change.

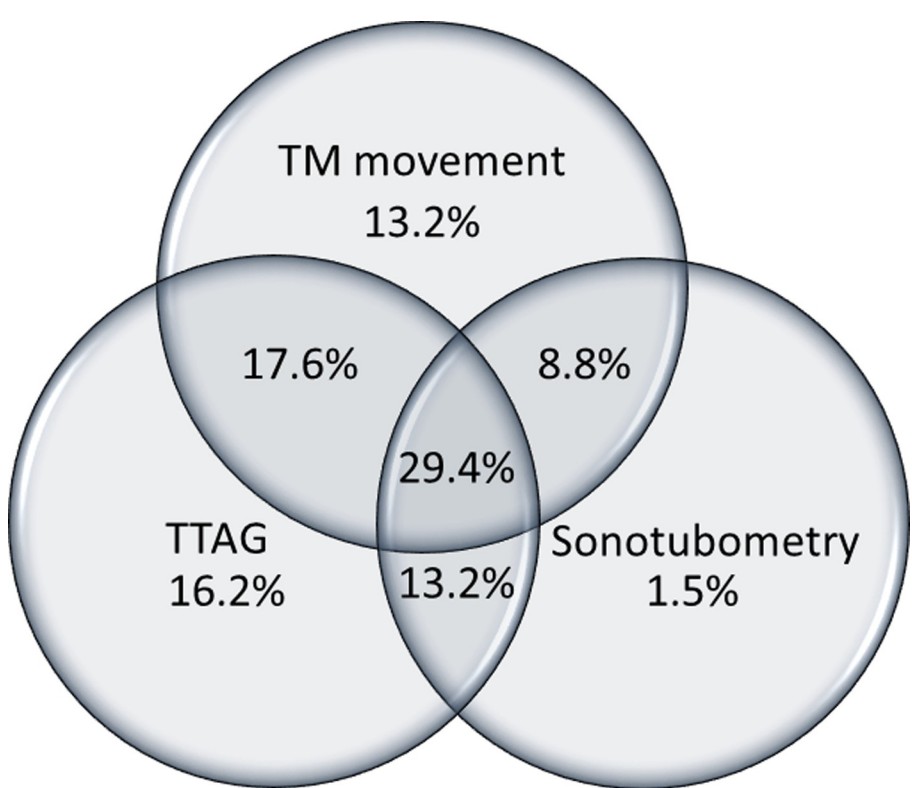

**Fig 5. A Venn diagram of TM movement, TTAG and sonotubometry.** A Venn diagram of three objective tests (TM movement, TTAG and sonotubometry) was drawn from 68 ears of "Definite PET" in which all the three tests were performed.

**Table 4. Clinical features of completely open and closed group according to CT.**

| | | CT completely open | | | CT closed | | | T test |
|---|---|---|---|---|---|---|---|---|
| | | Total | Positive | % | Total | Positive | % | |
| Diagnosis of PET at first consultation | | 48 | 48 | 100.0% | 22 | 22 | 100.0% | |
| Aural symptoms | | | | | | | | |
| | Voice autophony | 48 | 46 | 95.8% | 22 | 19 | 86.4% | 0.12 |
| | Aural fullness | 48 | 41 | 85.4% | 22 | 21 | 95.5% | 0.11 |
| | Breathing autophony | 48 | 43 | 89.6% | 22 | 14 | 63.6% | < 0.01 |
| Tubal obstruction procedures | | | | | | | | |
| | Posture change | 48 | 45 | 93.8% | 22 | 18 | 81.8% | 0.10 |
| | Pharyngeal orifice obstruction | 3 | 3 | 100.0% | 4 | 4 | 100.0% | |
| Objective findings of patent E-tube | | | | | | | | |
| | TM movement | 42 | 32 | 76.2% | 18 | 10 | 55.6% | 0.07 |
| | TTAG | 48 | 38 | 79.2% | 22 | 16 | 72.7% | 0.29 |
| | Sonotubometry | 48 | 34 | 70.8% | 22 | 4 | 18.2% | < 0.01 |
| | below 100dB | 48 | 31 | 64.6% | 22 | 0 | 0% | < 0.01 |
| | plateau type | 48 | 5 | 10.4% | 22 | 4 | 18.2% | 0.21 |
| Other objective findings of patent E-tube | | | | | | | | |
| | Ohta method | | | | | | | |
| | upper 10dB | 36 | 22 | 61.1% | 18 | 5 | 27.8% | < 0.01 |

## Subjective findings

Voice autophony, a sense of aural fullness, and breathing autophony were observed in 93.6%, 87.2%, 78.2% of ears, respectively. Our previous survey in 135 different cases of patients indicated those incidences as 90%, 84%, 65%, respectively. This previous study included both "Definite PET" and "Possible PET" patients. These results seem to suggest that "Definite PET" patients have a tendency to incur a higher ratio of breathing autophony.

In addition, PHI-10 was used for evaluation of subjective severity of PET. We have previously reported that this scoring system is suitable for evaluating severity of PET and the efficacy of treatment, provided that the diagnosis of PET is certain [12].

## Objective findings

In the JOS Diagnostic Criteria, respiratory fluctuation of the TM, TTAG and sonotubometry were recognized as objective findings of patent Eustachian tube. The former two tests are similar in that both tests monitor pressure transmission from the nasopharynx to the middle ear, by imposing pressure change through respiration or sniffing, and evaluate its effect on the middle ear pressure [8]. Sonotubometry evaluates sound transmission from the nasopharynx to the external auditory meatus [10, 19]. Previous study indicate that definite PET can be diagnosed if sound attenuation from the nostril to EAC is less than 100 dB [20]. It is enhanced in patients with PET, demonstrating lowering of the probe tone SPL or open plateau pattern. Positive findings of respiratory fluctuation of the TM was observed in 69.1% of ears, while that of TTAG in 76.9% of ears, and that of sonotubometry in 55.1% of ears. A positive ratio of TM observation and TTAG were higher than sonotubometry in this study. Similar results were obtained in our previous study where 72.6% were positive in TTAG, and 41.5% were positive in sonotubometry based on the JOS Diagnosis Criteria announced in 2012 [14], which is same as the current Diagnostic Criteria except that the latter added pharyngeal orifice obstruction treatment as a tubal obstruction procedure and the probe tone SPL less than 100 dB as a positive finding of sonotubometry. Moreover, a combination of respiratory fluctuation of the TM and TTAG can detect PET in 98.5% of ears (Fig 5). These results suggest that evaluation of pressure transmission such as TM observation and TTAG is more sensitive than that of sound transmission represented by sonotubometry. However, it does not disregard the usefulness of sonotubometry. Previous study has revealed that probe tone SPL of sonotubometry could be more useful than TTAG to predict the morphological severity of PET [9]. As such, all ears with a probe tone lowered to a level less than 100 dB SPL in sonotubometry were included in the CT completely open group in this study to corroborate the findings.

In this study, it is evident that ET testing apparatus is efficacious. However, if we solely depend on testing apparatus without observing the respiratory fluctuation of the TM, 13.2% of cases would have remained as "Possible PET" due to the lack of objective findings of a patent ET. This result suggests that observation of TM is indispensable for PET diagnosis.

The TTAG is widely used for PET diagnosis in Japan. However, there is little data supporting its use in English literature. Recently, Smith et al. investigated the diagnostic value of various tests for ET function and stated that TTAG is recommended for use both in intact and perforated TMs, as it was found to be comparison with TM observation, sonotubometry, impedance and tubomanometry in sensitivity, specificity and ease of use, albeit in 12 cases [21]. Our results highlighted the usefulness of TTAG. Although TTAG was performed with careful attention to exclude such artifacts, further studies to validate the accuracy of TTAG measurements are needed.

Recently, Ohta's method [11, 13], a modification of sonotubometry, performed during postural change from the forward-bending to the sitting positions was investigated. This method

is based on the fact that PET symptoms are usually relieved or resolved by postural change from sitting or standing to recumbent or head-down positions. If the positive findings of Otha's method are added to the sonotubometry results, positive finding in sonotubometry would have increased by 10 ears and the positive ratio of sonotubometry would rise from 55.1% to 67.9% in this study. This new method of sonotubometry could contribute to increasing the rate of accurate diagnosis especially in situations where TTAG is not available.

Sitting 3-D CT is useful in the diagnosis of PET as shown in earlier studies [14–16, 22, 23]. The completely open group, which is considered to be as infallible PET, was significantly higher than the other groups in terms of incidence of positive breathing autophony, positive sonotubometry and positive Ohta's method. The sitting 3-D CT of the temporal bone is very useful because it helps in the diagnosis for both PET and superior semicircular canal dehiscence syndrome (SCDS) in the same examination, since the two diseases are similar in symptoms and its differentiation is mandatory [6, 24, 25]. However, as the sitting 3-D CT has not been widely used in many clinics to date, it may be too early to discuss its inclusion into the diagnostic criteria for PET.

## Limitations of this study

The number of the patients in this study is relatively small to represent characteristics of PET. Moreover, patients with relatively severe PET visit our department to seek treatment including Kobayashi Plug insertion [26, 27] and injection of the ET orifice [28]. A multicenter study will be necessary to overcome these limitations.

## Conclusions

The characteristics of main symptoms and the efficiencies of various tests in PET diagnosis were analyzed based on data obtained from "Definite PET" patients diagnosed by the JOS Diagnostic Criteria. The greater the availability of tests to evaluate PET, the greater the opportunities to diagnose "Definite PET". In particular, tests measuring pressure transmission between the nasopharynx and middle ear, such as TM observation and TTAG, are more sensitive than sonotubometry measuring sound transmission.

## Supporting information

**S1 Table. Summary of the clinical features of patients (raw data).**
(DOCX)

**S2 Table. Clinical features of completely open and closed group according to CT (raw data).** 1. CT completely open, 2. CT closed.
(DOCX)

**S3 Table. Summary of results from TTAG, sonotubometry, Ohta method and PHI-10.**
(DOCX)

## Author Contributions

**Data curation:** Yoshinobu Kawamura, Ryoukichi Ikeda, Hiromitsu Miyazaki.

**Project administration:** Ryoukichi Ikeda, Toshimitsu Kobayashi.

**Supervision:** Tetsuaki Kawase, Yukio Katori, Toshimitsu Kobayashi.

**Writing – original draft:** Yoshinobu Kawamura, Ryoukichi Ikeda.

**Writing – review & editing:** Ryoukichi Ikeda, Toshiaki Kikuchi, Toshimitsu Kobayashi.

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
