## [Decision Letter · Decision Letter 0]

9 Oct 2019

PONE-D-19-22143

The characteristic of Patulous Eustachian Tube Patients diagnosed by the JOS Diagnostic Criteria

PLOS ONE

Dear Dr. Ikeda,

Thank you for submitting your manuscript to PLOS ONE. After careful consideration, we feel that it has merit but does not fully meet PLOS ONE’s publication criteria as it currently stands. Therefore, we invite you to submit a revised version of the manuscript that addresses the points raised during the review process.

Please address the reviewers' comments below, in particular with regard to prior studies looking at similar symptoms.

We would appreciate receiving your revised manuscript by Nov 23 2019 11:59PM. To enhance the reproducibility of your results, we recommend that if applicable you deposit your laboratory protocols in protocols.io, where a protocol can be assigned its own identifier (DOI) such that it can be cited independently in the future. For instructions see: http://journals.plos.org/plosone/s/submission-guidelines#loc-laboratory-protocols

We look forward to receiving your revised manuscript.

Kind regards,

Jennifer Alyono

Academic Editor

PLOS ONE

Journal Requirements:

2. In ethics statement in the manuscript and in the online submission form, please provide additional information about the patient records used in your retrospective study. Specifically, please ensure that you have discussed whether all data were fully anonymized before you accessed them and/or whether the IRB or ethics committee waived the requirement for informed consent. If patients provided informed written consent to have data from their medical records used in research, please include this information.

'This work was supported by JSPS KAKENHI Grant Number 18K09366 and the Project Promoting Clinical Trials for Development of New Drugs and Medical Devices (Japan Medical Association) from Japan Agency for Medical Research and development, AMED.

'No'

'No'

6. Please include your tables as part of your main manuscript and remove the individual files. Please note that supplementary tables (should remain/ be uploaded) as separate "supporting information" files

Additional Editor Comments (if provided):

Reviewers' comments:

Reviewer's Responses to Questions

**Comments to the Author**

1. Is the manuscript technically sound, and do the data support the conclusions?

Reviewer #1: Yes

Reviewer #2: Yes

2. Has the statistical analysis been performed appropriately and rigorously? 

Reviewer #1: Yes

Reviewer #2: Yes

3. Have the authors made all data underlying the findings in their manuscript fully available?

Reviewer #1: Yes

Reviewer #2: Yes

4. Is the manuscript presented in an intelligible fashion and written in standard English?

Reviewer #1: Yes

Reviewer #2: Yes

5. Review Comments to the Author

Reviewer #1: This study has merit for reporting on characteristic of PET patients as diagnosed using the Otological Society of Japan Diagnostic Criteria, which was not done before. However, a study done recently by Smith, 2018, albeit with a smaller number of subjects has a more detailed analysis of the diagnostic criteria used in their evaluation.

I believe figures 5,6,8,9 can be all combined into one with 4 subsection

Figure 2 and 3 can also be combined with subsections

Did your study have any inclusion/exclusion criteria?

Abstract page 3

Line 13 – define TTAG

Line 17 . In ninety-one percent of the ears… -> In 91% of the ears…

Line 18 ….lying / lordotic position… -> do you mean supine? (lordotic is not often used within English literature, please correct to supine within the manuscript)

Methods page 8

Line 14 Criteria 3C use -> Criteria 3C used

Page 9 Changes in probe tone SPL exceeding 10dB was regarded as

a positive finding of PET – was this something that you defined as significant or previously defined and accepted as a significant finding,

Line 16 define CT

Results

Page 12

Line 10 As the three ears exhibited both positive findings… what is meant by “three ears”

Page 13

It is a very interesting finding that 43 (58.1%) had severe handicap

Page 15

Line 13 please change annoying to bothersome

Page 16

Line 14 nostril to EACl - nostril to EAC

References

Please remove number 5 as it is repeated twice

Figure 4

Closed short is not very definite imaging, maybe adding a bracket to show where the region of closed portion of ET is being evaluated

Reviewer #2: Thank you for this interesting and timely review on an important patient condition. Your study is well designed to address the question at hand within the limitations you mention, namely the high likelihood of diagnosing PET in your study population based on the type of referrals your center receives. With that said, this paper contributes to our body of knowledge by detailing diagnostic techniques and criteria, and is especially valuable in the comparison of various diagnostic modalities.

Several comments and requested clarifications below could strengthen this paper.

Page 8, Line 4: How was 2B conductive specifically?

Page 8, Line 5,6: Presume that any movement of the TM is defined as positive? Please clarify.

Page 9, Line 9,10: Please clarify in the text what is being monitored at the level of the EAC (sound transmitted from the nasopharynx?)

Page 12, Line 10-12: Three ears were positive for both probe tone testing and an open plateau pattern?

Page 14, line 10-13: Any correlation between these delayed “Definite PET” and severity of symptoms? Perhaps less because they were intermittent?

Page 14, line 13: Consider, “PET symptoms ARE intermittent”.

Page 14, line 17: Consider, “out institute received.

Page 15, line 3: Consider a more formal term in lieu of “Annoying” when referring to PET symptoms.

Page 15, line 2-11: Largely restating results. Is there any significance or specific characteristics to the patients that did not improve with postural changes that clinicians should be monitoring for?

Page 16, line 14: EAC1?

6. PLOS authors have the option to publish the peer review history of their article (what does this mean?). If published, this will include your full peer review and any attached files.

Reviewer #1: No

Reviewer #2: Yes: Pedrom C. Sioshansi, MD

---

## [Author Response · Author response to Decision Letter 0]

28 Oct 2019

Response to Reviewers

Reviewer #1

We wish to express our deep appreciation to the reviewer.

I believe figures 5,6,8,9 can be all combined into one with 4 subsections

Response: 

We have modified as your suggestion.

Figure 2 and 3 can also be combined with subsections

Response: 

We have modified as your suggestion.

Did your study have any inclusion/exclusion criteria?

Response: All patient diagnosed by JOS criteria were included. There was not exclusion criteria.

Abstract page 3

Line 13 – define TTAG

Response: We have defined as follows “Tubo-Tympano-Aerodynamic Graphy (TTAG)”.

Line 17 . In ninety-one percent of the ears… -> In 91% of the ears…

Response: We have changed “In ninety-one percent of the ears” to “In 91% of the ears”.

Line 18 ….lying / lordotic position… -> do you mean supine? (lordotic is not often used within English literature, please correct to supine within the manuscript)

Response: “Lordotic” means bending forward.

We have modified in the text. 

Methods page 8

Line 14 Criteria 3C use -> Criteria 3C used

Response: We have corrected as your suggestion.

Page 9 Changes in probe tone SPL exceeding 10dB was regarded as

a positive finding of PET – was this something that you defined as significant or previously defined and accepted as a significant finding,

Response: There have been no report to define and accepted report as a significant findings. Our results could contribute to define the significant findings.

Line 16 define CT

Response: We have defined “CT” as “cone-beam computed tomography (CBCT)”.

Results

Page 12

Line 10 As the three ears exhibited both positive findings… what is meant by “three ears”

Response: “Three ears” mean that three ears were positive for both probe tone testing and an open plateau pattern.

We have corrected in the text.

Page 13

It is a very interesting finding that 43 (58.1%) had severe handicap

We wish to express our deep appreciation to the reviewer.

Page 15

Line 13 please change annoying to bothersome

Response: We have corrected as your suggestion.

Page 16

Line 14 nostril to EACl - nostril to EAC

Response: We have corrected as your suggestion.

References

Please remove number 5 as it is repeated twice

Response: We have deleted as your suggestion.

Figure 4

Closed short is not very definite imaging, maybe adding a bracket to show where the region of closed portion of ET is being evaluated

Response: We have modified in the figure.

Reviewer #2

We wish to express our deep appreciation to the reviewer.

Page 8, Line 4: How was 2B conductive specifically?

Response: For example, swab is placed or a small amount of gel is injected into pharyngeal orifice through nasal cavity. Improvement of aural symptoms is regarded as positive.

Page 8, Line 5,6: Presume that any movement of the TM is defined as positive? Please clarify.

Response: We have modified as your suggestion.

Page 9, Line 9,10: Please clarify in the text what is being monitored at the level of the EAC (sound transmitted from the nasopharynx?)

Response: We have added the sentence “Pressure changes in the external auditory canal (EAC) and the nasopharynx were simultaneously recorded using the manometry mode of the TTAG.”

Page 12, Line 10-12: Three ears were positive for both probe tone testing and an open plateau pattern?

Response: “three ears exhibited both positive findings” means “three ears were positive for both probe tone testing and an open plateau pattern.”

We have added the sentence “As three ears were positive for both probe tone testing and an open plateau pattern,”.

Page 14, line 10-13: Any correlation between these delayed “Definite PET” and severity of symptoms? Perhaps less because they were intermittent?

Response: PHI-10 score of initial“Definite PET” and delayed “Definite PET” are 26.6±11.5 and 16.7±9.77 (p = 0.054), respectively. Delayed type tended to less severity but not significant because of small number of delayed type.

Page 14, line 13: Consider, “PET symptoms ARE intermittent”.

Response: We have corrected as your suggestion.

Page 14, line 17: Consider, “out institute received.

Response: We have corrected as your suggestion.

Page 15, line 3: Consider a more formal term in lieu of “Annoying” when referring to PET symptoms.

Response: We have changed “annoying” to “bothersome” as your suggestion.

Page 15, line 2-11: Largely restating results. Is there any significance or specific characteristics to the patients that did not improve with postural changes that clinicians should be monitoring for?

Response: Thank you very much for useful comment. We also are interested in this point. There are only 4 cases in this study. Further study is needed to elucidate this issue.

Page 16, line 14: EAC1?

Response: We have corrected as your suggestion.

---

## [Editor Report · Decision Letter 1]

10 Dec 2019

The characteristic of patulous eustachian tube patients diagnosed by the JOS diagnostic criteria

PONE-D-19-22143R1

Dear Dr. Ikeda,

We are pleased to inform you that your manuscript has been judged scientifically suitable for publication and will be formally accepted for publication once it complies with all outstanding technical requirements.

With kind regards,

Jennifer Alyono

Academic Editor

PLOS ONE
---

## [Editor Report · Acceptance letter]

16 Dec 2019

PONE-D-19-22143R1 

The characteristic of patulous eustachian tube patients diagnosed by the JOS diagnostic criteria 

Dear Dr. Ikeda:

I am pleased to inform you that your manuscript has been deemed suitable for publication in PLOS ONE. Congratulations! Your manuscript is now with our production department. 

With kind regards,

on behalf of

Dr. Jennifer Alyono 

Academic Editor

PLOS ONE